# A Patient with a Small Deletion Affecting Only Exon 1-Intron 1 of the *NXF5* Gene: Potential Evidence Supporting Its Role in Neurodevelopmental Disorders

**DOI:** 10.3390/genes16050571

**Published:** 2025-05-13

**Authors:** Yessica Yesenia Tapia, Claudia Ciaccio, Merve Begüm Bacınoğlu, Stefano D’Arrigo, Francesca Luisa Sciacca

**Affiliations:** 1Laboratory of Cytogenetic, Service of Laboratory Medicine, Department of Diagnostic and Technology, Fondazione I. R. C. C. S. Istituto Neurologico Carlo Besta, 20133 Milan, Italy; yessica.tapia@istituto-besta.it; 2Unit of Infantile Neuropsychiatry, Fondazione I. R. C. C. S. Istituto Neurologico Carlo Besta, 20133 Milan, Italy; stefano.darrigo@istituto-besta.it; 3Unit of Laboratory Medicine, Laboratory of Clinical Pathology, Fondazione I. R. C. C. S. Istituto Neurologico Carlo Besta, 20133 Milan, Italy; merve.bacinoglu@istituto-besta.it

**Keywords:** *NXF5*, intellectual disability, autism spectrum disorders, microdeletion Xq22.1

## Abstract

Genetic studies have identified numerous candidate genes for neurodevelopmental disorders associated with intellectual disability (ID) and autism spectrum disorders (ASD). Some genetic anomalies are very rare or challenging to detect, making it essential to validate the presence of gene mutations or copy number variations in additional patients with similar clinical phenotypes. **Background/Objectives**: Case reports play a crucial role in this process by validating rare variants in phenotypically matched patients, shedding light on novel candidate genes linked to these disorders. **Methods**: Patients with ID and ASD underwent neurological examinations, brain magnetic resonance imaging (MRI), sleep and wake electroencephalogram (EEG), neuropsychological evaluations, and laboratory tests including molecular analysis for fragile-X syndrome and array comparative genomic hybridization (aCGH). **Results**: We observed a patient with ID and ASD who carried a microdeletion in Xq22.1 that affects only exon 1 and intron 1 of the Nuclear RNA Export Factor 5 (*NXF5*) gene. The patient’s phenotypic features overlap with those of the only four previously reported cases of variations involving the same gene. **Conclusions**: Our findings suggest that *NXF5* may play a role in neurodevelopmental disorders and should be considered in the spectrum of X-linked ID associated with ASD.

## 1. Introduction

Intellectual disability (ID) affects approximately 1% of individuals undergoing genetic testing [1]. Advances in technology have facilitated the identification of novel genetic causes of ID, both syndromic—characterized by distinctive facial features, morphological anomalies, multisystemic disorders, or other neurological conditions—and non-syndromic, where ID occurs as an isolated feature [2,3]. ID is more prevalent in males than females [1], with several ID-associated genes located on the X chromosome. Among these, recent studies have highlighted the role of genes involved in mRNA processing and metabolism.

Nuclear RNA Export Factor 5 (*NXF5*) is one of the clustered genes in the Xq22.1 region, alongside Nuclear RNA Export Factor 2 (*NXF2*), Nuclear RNA Export Factor 2B (*NXF2B*), Nuclear RNA Export Factor 4 (*NXF4*), and Nuclear RNA Export Factor 3 (*NXF3*). These genes play a critical role in mRNA export from the nucleus to the cytoplasm, a key process in mRNA metabolism. TAP/NXF1 binds RNA via its N-terminal domain, interacts with the p15/NST protein through its central region, and connects to the nuclear pore complex (NPC) via its C-terminal domain. The NXF5 protein has an affinity for RNA and p15/NXT but lacks part of the C-terminal domain required for binding nucleoporins [4]. NXF5 is located in the nucleus and the cytoplasm, in cell bodies and neurites of hippocampal neurons [4]. *NXF5* has 16 exons and encodes at least 5 isoforms generated through alternative splicing, which excludes exons 4 and 5 (*NXF5a-e*, GenBank AJ277654-AJ277658). Not all isoforms of the *NXF5* gene have been functionally characterized. While alternative splicing of the *NXF5* gene results in multiple transcript variants, many of these isoforms are predicted to undergo nonsense-mediated decay due to the presence of premature stop codons in their mRNA sequences. Consequently, these isoforms are unlikely to produce functional proteins in vivo (https://www.genecards.org/cgi-bin/carddisp.pl?gene=NXF5&utm_source, accessed on 5 May 2025). Although translation begins at exon 3, a breakpoint in intron 1 has been shown to abolish *NXF5* mRNA expression [4] completely. The involvement of *NXF5* copy number variants (CNVs) in X-linked ID has been previously suggested in four reported cases (Table 1).

The first reported case involved a pericentric inversion with a breakpoint in *NXF5* intron 1, leading to a complete loss of *NXF5* expression. The patient, an adult male, exhibited severe ID, short stature, generalized muscle wasting, and distinctive facial features, including hypertelorism, down-slanting palpebral fissures, large and low-set ears, and a wide mouth. He also presented with ASD traits, such as anxiety, avoidance of eye contact, absence of speech, episodes of uncontrolled laughter, and significant impairments in communication and social functioning [4,5].

A second male patient carried a 0.8 Mb duplication in Xq22, involving 15 genes, including *NXF5*. His clinical presentation included moderate ID, speech delay, and facial anomalies such as down-slanting palpebral fissures, large, low-set ears, a broad nasal bridge, a long philtrum, and a wide mouth with prominent incisors and diastema. Additionally, he displayed ASD-related behaviors, including hyperactivity, short attention span, stereotypic hand movements, and temper tantrums [6]. While it remains unclear whether this duplication alters *NXF5* expression, his phenotype shares common features with the previously described male patient.

The other two reported cases involved a mother and daughter carrying a 1.9 Mb deletion affecting the *NXF* gene cluster and other genes. The deletion was de novo in the mother and inherited by her daughter. X-inactivation analysis revealed a skewed pattern in the mother, preferentially inactivating the deleted chromosome, whereas the daughter exhibited a random inactivation pattern. Clinically, the mother presented with mild ID, short stature, brachycephaly, coarse facial features, epilepsy, and borderline personality disorder. The daughter, on the other hand, had severe ID, speech delay, microbrachycephaly, severe muscle hypotonia, and distinct somatic traits, including hypertrichosis, a coarse facial structure, a small forehead, thick lips, a smooth philtrum, and low-set ears. She also exhibited ASD-related features such as stereotypic movements and self-injurious behaviors [7].

Additional cases have been reported in PubMed, including both male and female patients with large duplications spanning Xq21q24. As expected, their clinical manifestations varied significantly. Among them, one female patient had a 17.6 Mb duplication of Xq22.1q24, involving multiple genes and beginning with *NXF5*, which was disrupted [8]. However, aside from ID, she did not exhibit clinical features overlapping with the previously described cases.

The Decipher database (www.deciphergenomics.org) lists three female patients (#305777, #264512, #268030) with large deletions of 4.36 Mb, 4.88 Mb, and 5.63 Mb, respectively. In one case (#305777), the patient also carried a likely pathogenic mutation and had a complex phenotype. Patient #264512 was reported to have ID, while no phenotype information was available for patient #268030. Here, we present the case of a child with developmental delay (DD) and ASD.

## 2. Materials and Methods

Our patient is the only child of healthy, non-consanguineous parents of Moroccan origin. The family history was unremarkable, except for a cousin of the maternal grandfather, living in Morocco and not genetically tested, who was reportedly affected by an unspecified ID. The pregnancy was uneventful, and the child was born at 40 weeks via spontaneous vaginal delivery. At birth, he weighed 2700 g (small for gestational age), measured 49 cm in length, and had a head circumference of 35 cm. His Apgar scores were 9 (1 min), 10 (5 min), with no signs of perinatal distress. Due to the low birth weight, a transfontanellar ultrasound yielded normal results. He was breastfed and weaned regularly without any feeding difficulties. Due to global development delay, he first came to our attention at the age of 3 years and 3 months (42 months). Comprehensive assessments were conducted, including general and neurological examinations (ocular lobes alignment, complete MOE, evaluation of the VII cranial nerve; evaluation of tongue midline in the oral cavity; evaluation of tone, trophism, and strength, reflexes test; autonomous walking, running, and climbing on the chair). The child was minimally engaged, showed brief eye contact, repetitive play, and repetitive hand movements. Expressive language was limited to vocalizations and cooing. He talked a lot to himself. For these features, ASD questionnaire ADI-r Toddler (ADI-R) was administered to the father of the proband. Brain magnetic resonance imaging (MRI), sleep and wake electroencephalogram (EEG) were also performed.

Laboratory tests included: urinary mucopolysaccharides (spectrophotometric dosage); urinary organic acids (gas chromatographic profile); plasma amino acids profile (High-Performance Liquid Chromatography). We performed also hematochemical analyses for routine hospital admission: complete blood count, leukocyte formula, blood urea, nitrogen, glucose, creatinine, sodium, potassium, chloride, ammonia, aspartate transaminase, alanine transaminase, γ glutamyl transferase, alkaline phosphatase, pseudocholinesterase, creatinphophokinase, C-reactive protein, total proteins, prothrombin time, activated partial thromboplastin time, fibrinogen, uric acid, thyroid stimulating hormone, triiodothyronine and thyroxine. On DNA extracted from peripheral blood (Gentra Kit (Qiagen, Hilden, Germany) we performed molecular analysis for Fragile X syndrome (amplification of the CGG repeat sequence of the FMR1 gene using the AmplideX FMR1 PCR Kit, Asuragen, Austin, TX, USA, and fragment separation by capillary electrophoresis), and a-CGH. A-CGH is a molecular cytogenetic technique used to analyze CNVs, including deletions and amplifications, in genomic DNA. This method compares a test sample against a reference normal sample using competitive fluorescence in situ hybridization on an oligonucleotide-spotted platform. A-CGH analysis was performed using the Oligo ISCA 180 K platform, which includes a research-validated collection of specific probes designed to reliably detect CNVs with high resolution in genomic regions associated with genetic disorders. The array was designed by Oxford Gene Technology (OGT, Begbroke, Oxford, UK) and manufactured by Agilent Technologies (Santa Clara, CA, USA). The patient’s DNA was hybridized with sex-matched reference DNA from pooled controls (Promega, Madison, WI, USA) following the manufacturer’s protocol. Hybridization was carried out using the MaiTai™ Hybridization System (SciGene Corporation, Sunnyvale, CA, USA). After 20 h, the CytoChip oligo array was washed and scanned using the InnoScan 710 Microarray Scanner (Innopsys, Carbonne, France). CNVs were identified based on fluorescence intensity imbalances, with amplifications appearing as green (Cy3) signals and deletions as red (Cy5) signals. Data were analyzed using CytoSure Interpret software 4.10.44 version (OGT). Clinical interpretation of the aCGH results was based on published literature and publicly available databases, including ClinGen, Ensembl, USBC, the Database of Genetic Variants, Decipher, and the Italian database of Troina. The analysis followed the European and International Cytogenetic Guidelines [9,10]. Genomic coordinates were referenced according to the February 2009 Human Genome Build (GRCh37/hg19).

## 3. Results

The clinical history reported delayed achievement of developmental milestones: the patient started crawling at 9 months, standing unsupported at 12 months, and taking his first steps at 20 months. Expressive language development was limited to vocalizations and jargon. The Griffiths-III Developmental Scales revealed moderate global DD, with a non-uniform General Quotient (GQ) of 31 and an age-equivalent score of 13 months. Specifically, gross motor skills were at a 36-month level, whereas language abilities were severely delayed, corresponding to a 7-month level. The ADI-R questionnaire confirmed the suspicion of ASD. General and neurological examinations were unremarkable: ocular lobes were in normal alignment, complete MOE, no deficit of the VII cranial nerve; tongue midline in the oral cavity; tone, trophism, and strength within normal limits; symmetric, lively, midline reflexes present; autonomous walking with no pathological findings; he runs and climbs easily on the chair.

At 4 years and 6 months, he was admitted to the inpatient clinic for further diagnostic evaluations. By that time, his growth parameters were at the upper limits of the normal range for height and weight (weight: 22 kg, 90th–97th percentile; height: 113 cm, 90th–97th percentile), but he presented with relative microcephaly (head circumference: 50 cm, 25th–50th percentile). Once again, general and neurological examinations were unremarkable. The child was found to have ASD with associated language impairment and moderate-to-severe global developmental delay, presenting a highly atypical functioning profile characterized by greater impairment in verbal and non-verbal communication skills, despite relatively preserved gross motor development. The clinical picture was characterized by severe impairment in communication, social, and adaptive skills, requiring significant support and continuous assistance not only for educational and teaching aspects but also for daily activities such as personal hygiene, dressing, and feeding, in which the individual remains dependent on the caregiver (severity level 3 according to DSM-5).

Brain magnetic resonance imaging (MRI), sleep and wake electroencephalogram (EEG), neuropsychological evaluations, and laboratory tests resulted normal or not clinical relevant; plasma aminoacid profile was characterized by concentrations of amino acids citrulline and arginine at the lower end of the reference range, likely related to the current nutritional regimen, while the other amino acids involved in the urea cycle are not altered). Thyroid-stimulating hormone was at the lower limit 0.20 (ref. range 0.45–3.50), but triiodothyronine and thyroxine were within normal limits. Also molecular analysis for Fragile X syndrome resulted normal.

aCGH identified a deletion (Figure 1) in the long arm of the X chromosome, specifically in the Xq22.1 region: [GRCh37] Xq22.1(101,109,070_101,111,498)×0. The proximal breakpoint was located between nucleotides 101,111,498 and 101,109,070, while the distal breakpoint was between 101,113,228 and 101,114,002, resulting in a deletion spanning approximately 1.7–4.9 Kb. This deletion led to the loss of exon 1 and part of intron 1 of the *NXF5* gene, rendering this region nullisomic. The patient’s mother was found to carry the same deletion in heterozygosity, but exhibited no apparent clinical signs of the disorder; no standardized test has been performed, but she was perfectly able to understand and communicate, and she has always been autonomous in routine activities.

## 4. Discussion

Here, we describe a patient carrying a deletion in the *NXF5* gene, which possibly determines the loss of gene product [4]. To date, only four similar cases with *NXF5*-involving CNVs in the Xq22 region have been reported; it is particularly noteworthy that the patient described here exhibits physical features (microcephaly and short stature) and neurological traits (language delay, ASD) similar to those previously reported in male patients. Among the two reported female patients, mother and daughter, only the daughter presented similar features; however, it is worth noting that the mother shows a skewed X-chromosome inactivation pattern, thus showing a not completely penetrant phenotype. The mother of our patient carried the same deletion as her son but was asymptomatic. She is heterozygous, and a skewed X-chromosome inactivation pattern could explain the lack of symptoms, as well as a possible recessive effect of this deletion, which is limited to exon1-intron1 of the *NXF5* gene. Loss of exon 1-intron 1 of the *NXF5* gene could disrupt mRNA export or splicing in several key ways: loss of the N-terminal domain of the protein, usually located in exon 1 (leucine-rich nuclear export signals, RNA recognition motifs) and involved in mRNA binding and exporting; aberrant transcription initiation leading to truncated or non-functional isoform; suppression of regulatory elements usually located in intron 1; disruption of pre-mRNA splicing and cause exon skipping, or alter exon junction complexes, which are important for downstream mRNA exporting. Furthermore, it has been previously reported that a breakpoint in intron 1 abolished NXF5 mRNA expression [4].

Although the size and type of CNVs (deletions or duplications) varied among the reported individuals, some common clinical features were observed, including moderate to severe ID, speech delay or absence, distinct facial characteristics, and ASD features. Microcephaly was a feature of our patient and has been reported in two other patients besides the present case, one male and one female. This could be a characteristic with incomplete penetrance.

*NXF5* belongs to a gene cluster located in Xq22.1 that encodes nuclear export-associated proteins, which regulate the export of mRNA from the nucleus to the cytoplasm. Additionally, the localization of NXF5 aggregates in the cell bodies and neurites of hippocampal neurons [4] suggests its potential role in mRNA metabolism within the cytoplasm and possibly in synaptic plasticity [11]. The involvement of RNA metabolism in cognitive impairment is not a new concept, as it has been well established in the case of the fragile X syndrome-associated protein FMRP, which is also implicated in RNA nuclear export [12].

The role of *NXF5* CNVs in X-linked intellectual disabilities (XLID), potentially with syndromic features, has been considered “highly questionable” [13] due to the scarcity of reported cases with ID. However, more recent studies have associated *NXF5* variants with ASD [14]. It is worth noting that the identification of mutations through high-throughput sequencing may be hindered by filtering processes that exclude certain genes from variant analysis. Additionally, non-coding regulatory regions—potentially critical for gene expression—are typically not sequenced.

*NXF5* variants have been reported in the “Exome Sequencing Project Exome Variant Servers (EVSs)” which includes individuals with cardiac, pulmonary, and metabolic conditions [13]: a potential association of these variants with neurological features cannot be ruled out. Neurological examinations are not systematically performed in this cohort, leaving open the possibility that some individuals may have unrecognized neurological comorbidities. Furthermore, an *NXF5* exon 7 mutation (R113W) has been identified in a large family with heart block disorder, affecting protein stability and localization [15]. Notably, no neurological abnormalities were reported in this family, but this finding does not preclude the potential pathogenicity of exon 1 deletions, which could affect gene expression. This hypothesis is endorsed by a study on a mouse model, showing that the *NXF7*, the mouse counterpart of human *NXF5*, plays a determinant role in RNA metabolism in the brain and affects some brain functions [16]. *NXF7* knockout mice exhibited altered social behavior and impairments in spatial–cognitive performance, likely due to dysfunction of synaptic plasticity in the hippocampus [16].

Another important consideration about the paucity of *NXF5* CNVs reported in patients with ID is that there is a limit in the detection of small variants using lower-resolution aCGH platforms [17] and variability in CNV interpretation and reporting [18] may contribute to underdiagnoses.

## 5. Conclusions

Given these considerations, this report highlights the potential role of *NXF5* CNVs as a candidate factor in XLID with ASD features, warranting further investigation, also regarding the sex-specific expression of the neurodevelopmental disorder.

The post-pandemic era continues to fuel advancements in biomedical science, especially with an increasing focus on personalized medicine, digital health, and interdisciplinary collaboration. The influence of artificial intelligence (AI) has grown significantly, especially during and after the pandemic. The combination of diverse databases, advanced algorithms, and omics technologies has greatly accelerated the incorporation of AI into various aspects of human life. These trends are reshaping how biomedical research is conducted and how healthcare is delivered, offering new hope for treating diseases and improving overall public health outcomes [19].

## Figures and Tables

**Figure 1 genes-16-00571-f001:**
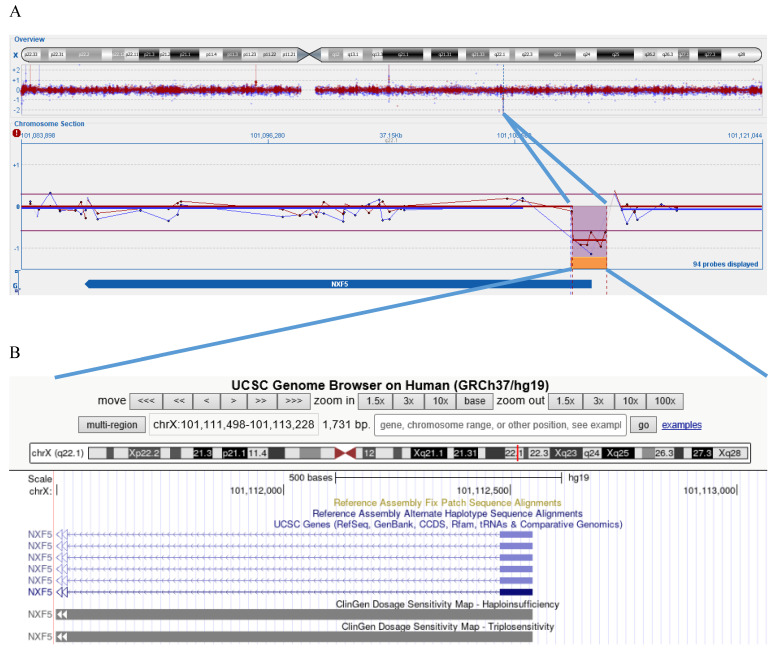
Panel (**A**) shows an overview of chromosome X and the region Xq22.1 involved in the imbalance of the patient (blue track) and his mother (red track); the purple lines highlight the magnifications of the deleted region (orange box). The tracks are reported and maximized in the lower chromosome section, where the position of *NXF5* gene is also shown. Panel (**B**) shows the microdeleted region in the UCSC Genome Browser, indicating that exon 1 and intron 1 of *NXF5* gene are deleted in the patient, resulting in nullisomy of this region, and for the mother, resulting in monosomy.

**Table 1 genes-16-00571-t001:** Review of previous reported cases carrying imbalances involving *NXF5* gene. ID: intellectual disability; ASD: autism spectrum disorder; M: male; F: female; FISH: Fluorescent In Situ Hybridization; ADHD: Attention Deficit Hyperactivity Disorder; mat: maternal; aCGH: array Comparative Genomic Hybridization; dn: de novo.

Patients Published in	Sex	ID	Peculiar Characteristics	Body Parameters	Language	ASD	Other Features	Unbalance and Method
[5]	M	Moderate–severe	Present	Short stature;	Absent	Present	Muscle waste	Inversion
				low weight;				FISH
				microcephaly				Breakpoint in *NXF5*
[6]	M	Moderate	Present	Normal	Dealy	Present	ADHD	Microduplication involving 15 genes
							hypotonia	aCGH
[7]	F	Mild	Present	Short stature	Normal	No	Borderline personality disorder;	Microdeletion, de novo, involving 15 genes;
							epilepsy	X-inactivation skewed 90:10
[7]	F	Severe	Present	Short stature;	Absent	Present	Hypotonia;	Microdeletion, mat, involving 15 genes;
				microcephaly			scoliosis	X-inactivation random

## Data Availability

Dataset available on Decipher database (ID 480203).

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
