# Peer review of "A Patient with a Small Deletion Affecting Only Exon 1-Intron 1 of the NXF5 Gene: Potential Evidence Supporting Its Role in Neurodevelopmental Disorders"

_genes, 2025, doi:10.3390/genes16050571_

Round 1
Reviewer 1 Report
Comments and Suggestions for Authors
This case report and literature review present a patient with a small deletion affecting exon 1-intron 1 of the NXF5 gene, contributing to the growing evidence of NXF5's role in neurodevelopmental disorders (NDDs), particularly intellectual disability (ID) and autism spectrum disorder (ASD). The authors provide a thorough clinical and genetic characterization of the patient, alongside a comparative analysis of four previously reported cases with NXF5 copy number variations (CNVs). The study underscores the potential pathogenicity of NXF5 disruptions and advocates for its inclusion in the diagnostic workup of X-linked ID/ASD. While the manuscript is well-structured and clinically relevant, minor revisions—particularly in clarity, discussion depth, and referencing—are recommended to strengthen its impact.
Abstract:
"The search for genetic causes [...] is uncovering a vast number of candidate genes" (line 5–6): Consider rephrasing for conciseness, e.g., "Genetic studies have identified numerous candidate genes for NDDs."
"Case reports play a crucial role in this process" (line 9): Specify how they contribute, e.g., "by validating rare variants in phenotypically matched patients."
Introduction:
"ID is more prevalent in males" (line 34): Cite a meta-analysis to support this claim.
"NXF5 has 16 exons and encodes at least five isoforms" (line 44–45): Clarify whether all isoforms are functionally characterized or hypothetical.
Case Presentation:
"His Apgar scores were 9 at 1 minute and 10 at 5 minutes" (page 4): Standardize formatting (e.g., "Apgar scores: 9 (1 min), 10 (5 min)").
*"Griffiths-III Developmental Scales revealed [...] GQ 31"* (page 4): Define "GQ" (General Quotient) at first mention.
Results:
The genomic coordinates of the deletion (page 5, line 151–154) are unclear due to line breaks. Revise for readability, e.g., "Xq22.1(101111498x1, 101109070_101111498x0, 101114002x1)."
Discussion:
Mechanistic Insights: Expand on how NXF5 loss (exon 1/intron 1) may disrupt mRNA export or splicing
Mouse Model Data: Highlight the NXF7 knockout mouse to strengthen the translational relevance of NXF5 in synaptic plasticity.
Clinical Correlations:
Compare the patient’s microcephaly and speech delay with prior cases (Table 1) to emphasize phenotypic consistency. Note exceptions (e.g., the female with Xq21q24 duplication lacking ASD).
Address the mother’s lack of symptoms despite heterozygosity: Could skewed X-inactivation (as in Grillo et al.’s case) explain this?
Technical Considerations:
Clarify why aCGH was chosen over higher-resolution methods on small CNV detection limits.
Discuss some new technology for future, cite paper “Integration in Biomedical Science 2024: Emerging Trends in the Post-Pandemic Era”
Comments on the Quality of English Language
ok
Reviewer 2 Report
Comments and Suggestions for Authors
The manuscript presents a case study of a patient with a small deletion affecting exon 1-intron 1 of the NXF5 gene. It explores its potential role in a developmental disorder. While the study may interest researchers focused on rare genetic variants, the manuscript lacks sufficient data. It has several weaknesses that limit its scientific rigor.
Major Concerns:
- Missing Laboratory Data- The manuscript mentions that laboratory tests were conducted. However, these results are not included in the case report. Providing these details would strengthen the clinical relevance of the findings.
- Insufficient Comparison to Prior Cases- The authors state that they reviewed clinical characteristics of previously reported patients with NXF5-related copy number variations (CNVs). However, they do not provide detailed comparisons or references. A more thorough discussion of prior cases would help contextualize the current findings.
- Lack of Detailed Neurological Examination- If available, including detailed neurological examination results would enhance this case study's clinical and scientific impact.
- Unclear ASD Features- The discussion concludes a possible association with autism spectrum disorder (ASD). However, the manuscript does not clearly describe the ASD-related features in this patient. Clarifying these observations is essential to support the authors' conclusions.
Round 2
Reviewer 2 Report
Comments and Suggestions for Authors
The authors have addressed my previous concerns and do not have further comments on the manuscript.